# The Transformation and Development Strategy of Waterside Villages through Transport System Reconstruction: A Case Study of Anxin County, Hebei Province, China

Chaoqun Wang and Jie He *

School of Architecture, Harbin Institute of Technology (Shenzhen), Shenzhen 518055, China;
15702414312@163.com
* Correspondence: hejie2021@hit.edu.cn

**Abstract:** The main method of transportation of waterside villages has changed from water to land transportation because of water conservation policies, dried-up rivers, or other reasons around the Baiyangdian Lake area. To guide waterside villages around Baiyangdian Lake to adapt to modern transport systems and deal with the urbanization waves in China, this study first measured temporal accessibility and potential change under land transportation by spatial centrality indices at three different points in time (1964, 1996, and 2008) from the historical road system we reconstructed. Then, based on these indices, we proposed a village structure for decision-making support. The results show that (1) the connectivity between the road network and water in Anxin County was weakened from 1964–2008. (2) Villages with high accessibility have changed from relying on water to clustering, homogeneity, and following main highways. (3) Villages with high potential have changed from meeting the previous conditions of being close to water or main highways to having both main roads and a cluster of other villages in the vicinity. (4) Anxin County's waterside villages can be divided into core villages, sub-core villages, connectivity villages, and sub-villages. The spatial structure formed by these four types is not only adapted to the modern transport system but also can serve the purpose of land consolidation or residential mobility focused by local government.

**Keywords:** waterside villages; transport infrastructure; rural development; remote sensing; network analysis; sustainable development

## 1. Introduction

Rapid urbanization has placed substantial pressure on China's rural areas and contributed to their spatial reconstruction and transformation. New transport infrastructure, like highways, railways, has played a major role in the urbanization of villages [1–6], especially waterside villages [7], which formerly relied on traditional water transportation. In historical geography, the term "waterside villages" refers to a specific geospatial area where settlements have long been influenced by a nearby body of water in terms of their location, formation, and evolution [8]. However, this has now changed as land transportation infrastructure has gradually improved and become more convenient than water transportation. A modern transport system, which is composed of strong land transportation and weak water transportation, is emerging. We argued that, for those traditional waterside villages, in order to deal with the wave of urbanization, there is an urgent need to find a strategy to adapt to modern transport systems.

Most studies of waterside villages have focused on how villages have adapted to being located near the water or on their spatial distribution characteristics [9–17], rather than examining how transportation mode shifting happened and what consequences it has caused in the region. Previous research established the double effects of land and water transportation modes. Water shaped the original spatial morphology and structure of settlements near bodies of water, while modern land transportation has drastically reshaped

them in the last century. For example, the waterborne-transportation-dominated structures and corresponding spatial layouts of the villages in the Lixia River region of northern Jiangsu Province (in east China) have been rapidly reconfigured by the strong intervention of the land transport system since the end of 19 century. Settlement patterns in this region shifted the original agglomeration in the south, with a dense water network, to the north, with a dense transportation network. Administrative villages and commercial land use also began to be distributed along main roads and were no longer gathered near the waterways or wharves [7]. In addition, there are also examples of the polarizing development results of the water and land transportation. Some villages in the upper reaches of the Erhai Basin in Yunnan Province (a border province in southeast China), which are generally limited by high ecological sensibilities of water body, developed slowly; however, other villages in this area developed rapidly thanks to improved road networks and other resources [3]. The number and area of waterside villages in the plains region, such as the Jianghan (whose name literally means "Hanjiang River and Yangtze River") Plain region and Gong'an County of Hubei Province (in central China), show a dual orientation toward roads and water transportation, although they are more significantly correlated with roads [1,4]. Roads have stimulated villages along the route to progress rapidly by accelerating the flow of commodities, people, capital, and technology [10,18]. In contrast, water-based transportation no longer has much impact because of conservation policies, dried-up rivers, or other reasons. In summary, the above studies further illustrate that for waterside villages with fading water transportation, developing land transportation seems a cost-effective way for promoting urbanization and stimulating rural revitalization.

About 50% of Anxin County, in Xiong'an New District, is covered by Baiyangdian Lake, the largest lake in the North China Plain and a feature that led to dramatic changes in transportation modes in the past. Before the 1960s, Baiyangdian Lake was the transportation hub of the Daqing River system. After the 1960s, due to the construction of upstream and downstream dams, all external waterways were cut off and only internal waterways were retained [19]. During the lengthy drought of 1983–1988, the government had to use "water diversions" to secure the water supply of Baiyangdian Lake. Both the loss of external waterways and the water shortage led to a decline in water transportation. At the same time, land transportation developed rapidly. By 2008, a comprehensive road network was established. In general, the transportation mode in Anxin County shifted from water-based to land-based transportation from the 1960s to the early 2000s, until land-based transportation became the mainstay and internal water transportation secondary. Today, the construction of an efficient land transportation system is the main objective of the Xiong'an New District plan [20] and regulations have been introduced to shift the center of gravity of the population in the area further inland [21]. There is an urgent need for a new land transportation-oriented development strategy to guide the upcoming modern transport system transformation, residential mobility, and land consolidation in Anxin County.

A land transportation-oriented spatial organization model is thus needed, but this has been lacking in previous studies. Therefore, we take the waterside villages in Anxin County as an empirical case to explore the following questions: (1) How has the land transportation accessibility and potential energy of waterside villages changed under the transportation mode shifting? (2) In what ways can villages respond to this shift? To answer these questions, we focus on two attributes related to land transportation. The first is *accessibility*, which refers to the ability of a normal village to be reached in the road network. The second is potential energy, simply called *potential*, which is defined as the capability of a normal village to develop into an important node in the road network. *Potential* can be considered as a comprehensive evaluation of accessibility and socio-economic condition. Although the detailed relationship between the *accessibility* of villages and the villages' socio-economic conditions is worth being considered, correlation and regression analysis in our pre-experiment indicated that correlation was not significant before modern land transportation system establishment and only weakly significant afterward. The analytical

result also implicates that the reason for villages' development was much more complicated than a single reason in land transport location. Therefore, this study only provides a quantitative evaluation of the two attributes, *accessibility* and *potential*. We further proposed a village structure to adapt to the land transport system based on its potential.

## 2. Materials and Methods

### 2.1. Study Area

Anxin County is part of Baoding City, Hebei Province, China. It is bordered by Renqiu County to the east, Gaoyang County to the south, and the Qingyuan and Xushui counties of Baoding City to the west (Figure 1a) and was jointly designated as the Xiong'an New District (the newest state-level economic zone in China) with Xiong County and Rongcheng County on 1 April 2017. Anxin County lies between latitudes 38°10′ and 40°00′ north and longitudes 113°40′ and 116°20′ east and has a total area of 738.6 square kilometers. There are 12 towns and 207 administrative villages under its jurisdiction. Baiyangdian Lake comprises 50% of Anxin County.

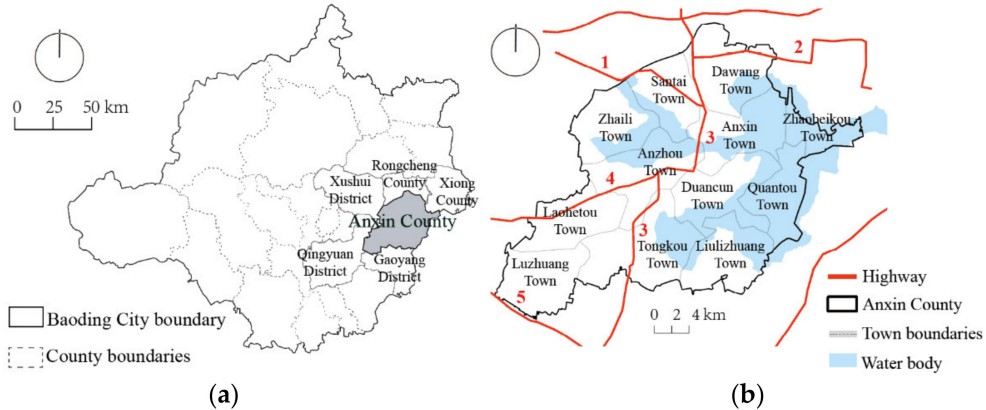

(a)  (b)

**Figure 1.** Study area: (**a**) Map of the study area; (**b**) Administrative divisions and main highways in Anxin County; 1: Xushui-Anxin County Highway; 2: Baojing Provincial Highway; 3: Rongcheng-Lixian National Highway; 4: Baoding-Anxin Provincial Highway; 5: Gaoyang-Baoding Provincial Highway.

Anxin County currently relies mainly on land transportation, supplemented by water transportation. There are four major highways in the county (Figure 1b). The Rongcheng-Lixian Provincial Highway (S235) crosses over Anxin County from north to south to connect Rongcheng County, Anxin County, and Gaoyang County. The Baojing Provincial Highway (S334) intersects with the Rongcheng-Lixian Provincial Highway on the northern shore of Baiyangdian Lake, leading to Xiong County in the east. The Baoding-Anxin Provincial Highway (S236), which leads to downtown Baoding City in the west from the Rongcheng-Lixian Highway in the center of Anxin County, was upgraded from a county highway to provincial level in 2002. The Xushui-Anxin County Highway, linking the Rongcheng-Lixian Provincial Highway in the downtown of Anxin and connecting the Xushui District of Baoding City to the west, was rerouted from Anxin County to Shanxi Village in 2003. The Gaoyang-Baoding Provincial Highway (S331), located along the southern border of the county and connecting Baoding City and Gaoyang County, also contributes to the transportation network.

### 2.2. Materials

Three significant years were selected for examining the changes in land transportation's *accessibility* and *potential*. In 1964, water transportation in Anxin County began to decline. In 1996, the modern road network construction began and was essentially completed in 2008.

Data for 1964 are from declassified CORONA satellite images from September 21, 1964. Data for 1996 and 2008 are derived from Landsat satellite image acquisitions from 13 May 1996 and 21 June 2008. Because of the long time interval between 1964 and 1996, we also checked remote sensing (RS) images acquired on 29 May 1972 as a reference for this interval but do not include them in the subsequent centrality indices' calculation because of the insignificant differences from 1964. To calibrate RS images and scanned maps, Landsat OLI images from 9 May 2020 were selected as the standard base map. All RS images and historical maps were projected in the WGS84 coordinate system UTM50N projection. These image data and their parameters are shown in Table 1.

The spatial data in this study can be divided into two categories. The first is point data representing villages (Figure 2). After interpreting settlement patches from historical RS images and performing collation and segmentation according to the village directory in the county annals [22], we used ArcGIS to convert these vector polygon features into point data. To calculate the weight of each village, quantitative data included population and per capita income derived from statistics in the county annals [22,23] were assigned as attributes of the village point. The coverage areas of the villages' patches in the study years were automatically generated as attributes of the village points. In terms of geographical characteristics, the villages can be roughly divided into three types: water villages entirely surrounded by water, land villages located entirely on land, and semi-waterside villages located in water-land border areas. We only selected the latter two to build the village dataset, because they would be more affected by land transportation than those entirely on the water. The study also incorporates villages outside the county boundary into the original dataset to overcome the edge effect in spatial analysis. In total, 264 villages were included. The second category is the road network, which is represented as line data. Dikes were also digitalized and added to this dataset because they also serve as transportation routes. The road network is mainly derived from the vectorization of administrative maps in county annals of the studied year and supplemented by manual interpretation from remote sensing images.

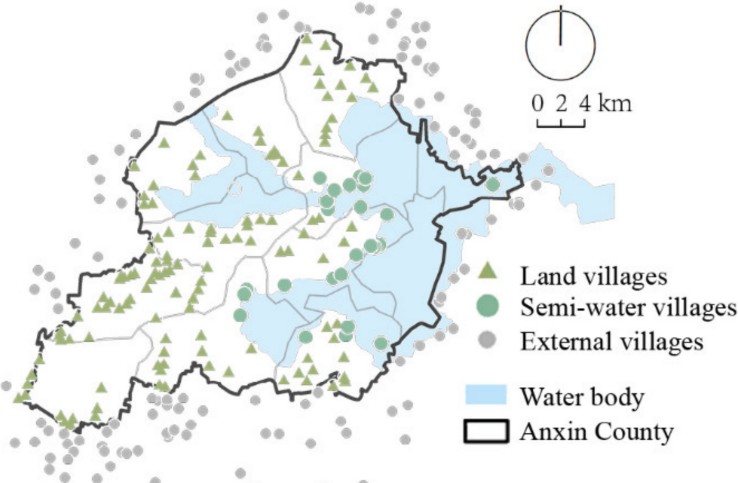

**Figure 2.** Point data of villages.

**Table 1.** Image materials used by study.

| Year | Map Materials | Source | Scale/Resolution | Coordinate System |
|---|---|---|---|---|
| 1964 |  | Declassified Corona Satellite Imagery | 3 m × 3 m | None |
| 1972 |  | Declassified Corona Satellite Imagery | 3 m × 3 m | None |
| 1996 |  | Anxin County Chronicle | 1:50,000 | None |
| |  | Landsat TM | 30 m × 30 m | WGS84 UTM50N |
| 2008 |  | Anxin County Chronicle 1978–2008 | 1:50,000 | None |
| |  | Landsat ETM | 30 m × 30 m | WGS84 UTM50N |
| 2020 |  | Landsat OLI | 30 m × 30 m | WGS84 UTM50N |

### *2.3. Methodology*

In this study, we measured *accessibility* by calculating road network centrality indices and evaluated *potential* by calculating weighted network centrality indices and the weight calculated from population and per capita income data. For each measurement, multiple indices were dimensionally reduced by factor analysis.

#### 2.3.1. Spatial Network Structure

Regarding the spatial accessibility of villages, there have been a number of studies using Euclidean distance to assess the accessibility of transport infrastructure, regardless the topological structure of the road network. In this research, we transformed the road layers to network datasets in ArcGIS and exported drive time and length attributes as impedance for spatial network analysis. We considered both their levels, as recorded in the county annals [22,23], and pavement material of different ages, then estimated the travelling speed on them, as shown in Table 2. For each road, the drive time attribute is equal to length divided by the four speed levels in 90 (level 1), 60 (level 2), 40 (level 3), and 20 (level 4) km/h.

**Table 2.** Road level definitions and speed estimates.

| Year | Level |
|------|-------|
| 1964 and 1972 | County Highway: 2; Other: 4 |
| 1996 and 2008 | Provincial Highway and County Highway: 1; Township Road: 2; Village Road: 3; Other: 4 |

#### 2.3.2. Multiple Centrality Assessment

We used multiple centrality assessment (MCA) in this research to provide a multi-faceted assessment of the importance of a village to the road network by measuring a range of centrality parameters. MCA is applied in urban planning, economic geography, and other fields to explore the link between urban activities or phenomena and transportation. Several studies have shown that multiple centrality indices can closely delineate the geographic characteristics of road networks by reflecting the advantages of different forms of accessibility [24]. These indices have both internal correlation and differences [25] and are often used in exploring the correlation with some geographical features and spatial activities [24–30]. We abstract the real road network by defining roads as links in a network and road intersections as nodes. All distance measurements are anchored along the real road network [24].

In this study, the calculation of multiple centrality indices is carried out using a toolbox for the ArcGIS Network Analyst extension named Urban Network Analysis (UNA), which was developed by Andres Sevtsuk of MIT [31]. UNA generates five parameters: reach, betweenness, closeness, gravity, and straightness. Reach measures how many surrounding points each point on the network reaches within a given search radius; betweenness is the fraction of the shortest paths between pairs of points within the search radius that passes through a given point; closeness is defined as the inverse of the cumulative distance required to reach from a given point to all other points in the network that fall within the search radius; gravity measures attraction to nearby villages, measured by the spatial impedance required to travel to each of the other points within the search radius; and straightness measures how many of the shortest paths from a given node to all other nodes within the search area are straight Euclidian paths.

All the parameters can be weighted according to the particular characteristics of nodes. The search radius is designated as travel times by car of 5, 10 and 60 min to study the road centrality at different scales. The former two thresholds are equivalent to pedestrian networks of 15 and 30 min. Fifteen minutes is the ordinary time limit of walking distance applied in planning, and a thirty minute threshold is in accordance the maximum radius for walking to farm work. They were used to examine the local network. Moreover, our

pre-experiments indicated that most villages can be reached within 60 min and there are gradients of their accessibility results. So, 60 min are suitable for examining the global network. Additionally, the straightness centrality search radius can only be designated as a distance, 1500 and 5000 m are used. The calculated parameters are given in Table 3.

**Table 3.** Indices to be calculated and their abbreviations.

| Index | Abbreviation |
|---|---|
| Global Reach | GR |
| Global Betweenness | GB |
| Global Closeness | GC |
| Global Gravity | GG |
| Global Straightness | GS |
| 5 min (minutes) Local Reach | CR I |
| 5 min Local Betweenness | CB I |
| 5 min Local Closeness | CC I |
| 5 min Local Gravity | CG I |
| 10 min (minutes) Local Reach | CR II |
| 10 min Local Betweenness | CB II |
| 10 min Local Closeness | CC II |
| 10 min Local Gravity | CG II |
| Local Straightness | CS |

### 2.3.3. Factor Analysis

To measure villages' two attributes, which we mentioned in the introduction section, *accessibility* and *potential*, three indices are dimensionally reduced using factor analysis (Figure 3). First, non-weighted multiple centrality indices are reduced as the Comprehensive Transport Index to evaluate the transportation accessibility. Second, because the *potential* is a comprehensive evaluation of accessibility and socio-economic condition, the Village Development Index, downscaled by population, per capita income, area, and area growth, was used to measure socio-economic condition and served as the weight of village nodes. However, for 1964, due to a lack of data, the Village Development Index was directly normalized by the area of the residential patch. Finally, the Transport Development Index was extracted from the centrality indices weighted by the Village Development Index to examine the potential.

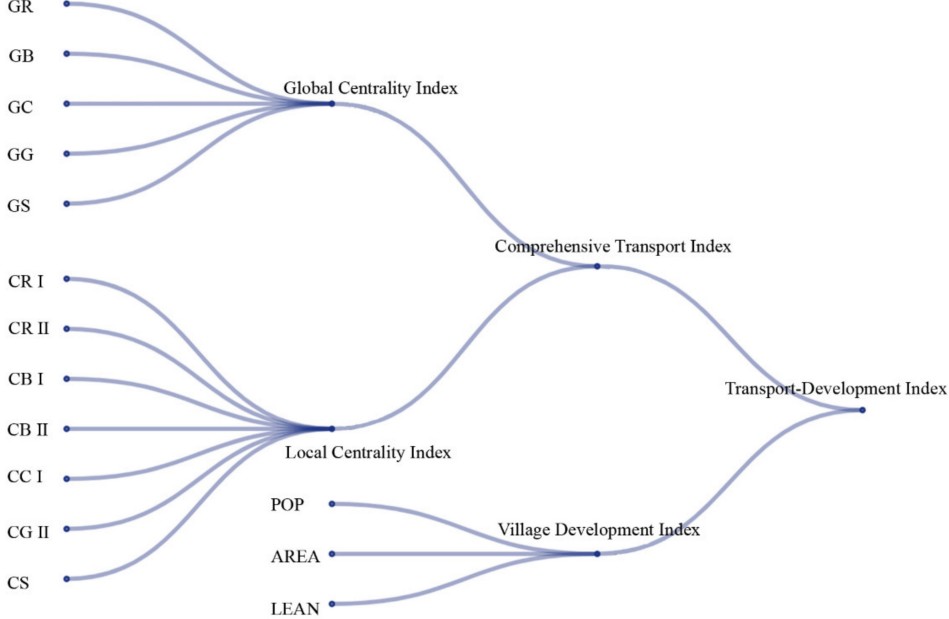

**Figure 3.** Factor analysis structure of multi-indices.

## 3. Results

### 3.1. Land Transportation System Change

The evolution of the land transportation system in the study area is illustrated in Figure 4. From 1964 to 1975, the road network in Anxin County was a jumble. However, it was already framed by Rongcheng-Lixian Provincial Highway, Baoding-Anxin Provincial Highway, and Xushui-Anxin County Highway. Many pathways led to main highways and the water body.

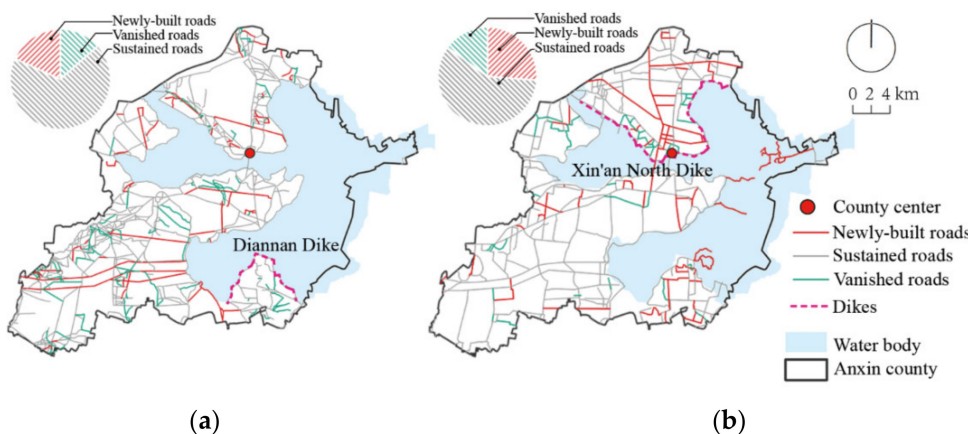

|          |          |
| :------: | :------: |
| (**a**)  | (**b**)  |

**Figure 4.** Changes in the road system of Anxin County between (**a**) 1964 and 1975 and (**b**) 1996 and 2008.

From 1975 to 1996, there were dramatic changes in the road network, especially in the villages south of the Diannan Dike, such as Liulizhuang Town. This area was prone to flooding before the dam was built in the 1970s, therefore making it more difficult to form a stable road network. By 1996, a land transportation system radiating to all villages had been established. The road network shape changed from chaotic to regular, and the clustering of pathways was no longer obvious.

From 1996 to 2008, there were no obvious changes in the road networks. Among the main highways, Rongcheng-Lixian Provincial Highway (northern section) and the Baojing Provincial Highway were completed. The other new roads that were added and revamped in this period were concentrated in the area of the county north of the Xin'an North Dike, the connection between waterside villages and the dike, and the area in Liulizhuang Town.

### 3.2. Centrality Indices

To detect changes in land transportation accessibility for villages, the centrality index in each study time was compared at the global and local scales. The temporal characteristics of Comprehensive Transport Index were then observed.

#### 3.2.1. Global Centrality Results

Figure 5 presents the results for global reach (GR), which represents the number of accessible villages for each origin. An increasing trend is found in most of the villages. However, Zhaili Town in the north and Liulizhuang Town in the south all maintained a low reach centrality across the years. In contrast, two other cases with similar locations—Dawang Town, which represents the only village cluster on the north border other than Zhaili Town, and Tongkou Town, just next to Liulizhuang Town—demonstrated rapid development from 1964 to 1996, as in 1996 they benefited from the construction of the Rongcheng-Lixian National Highway in the north and Gaoyang-Baoding Provincial Highway in the south, adjoining the county border. In this period, villages' global reach value in the south grew faster than in the north. However, from 1996 to 2008, this trend was reversed because of extensions to the road network around the county, as can be seen in Figure 4b.

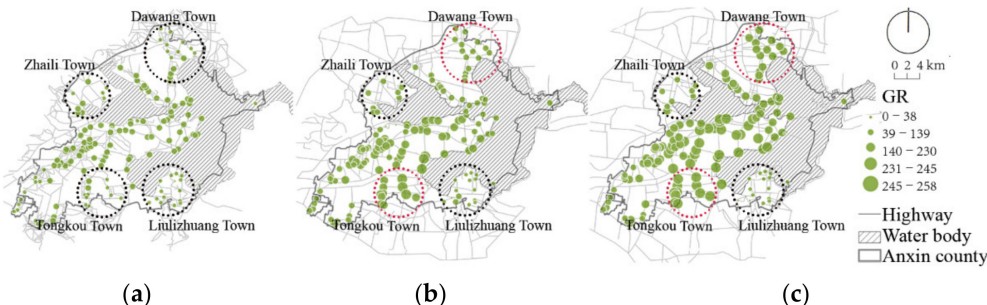

**Figure 5.** Global reach centrality (GR) of villages in (**a**) 1964; (**b**) 1996; (**c**) 2008.

Figure 6 presents the results for global betweenness (GB), indicating the likelihood of the village being passed through in the road system. In 1964, the intermediation role of semi-waterside villages was strong, but this advantage gradually disappeared from 1996 to 2008. In 1996, the villages on the north and south edge and along the Baoding-Anxin Provincial Highway had relatively strong intermediation, and this trend became pronounced in 2008. The villages around the county center had a strong intermediation role due to the extension of the road network in 2008.

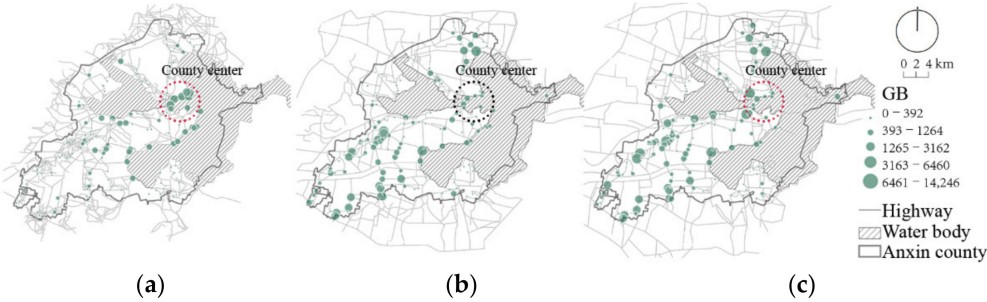

**Figure 6.** Global betweenness centrality (GB) of villages in (**a**) 1964, (**b**) 1996, and (**c**) 2008.

Figure 7 presents the results for global closeness (GC), indicating the geographical proximity of a village to other villages within a certain range. The closeness centrality results show opposite characteristics to those of the other parameters. Villages in Liulizhuang Town, which maintained a low value for other centrality metrics, had high closeness with neighboring villages. This suggests that this town had strong internal cohesion but a weak correlation with other towns. However, most other villages maintained a low closeness value.

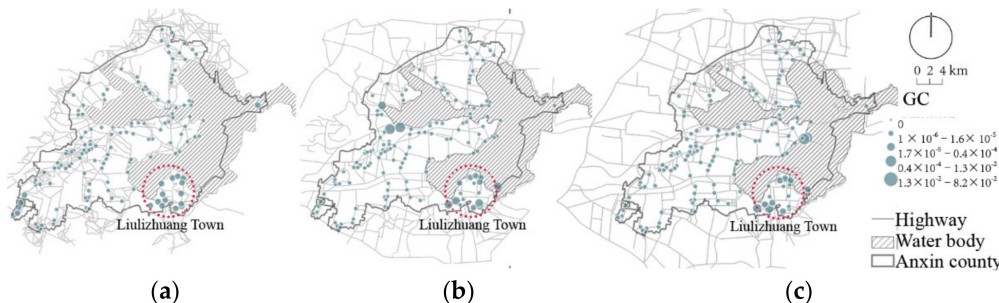

**Figure 7.** Global closeness centrality (GC) of villages in (**a**) 1964, (**b**) 1996, and (**c**) 2008.

Figure 8 shows the results for global gravity (GG), indicating the gravitational pull of a village on other villages within a certain range. High-GG villages that appear in clusters show strong interaction forces. In 1964, high-GG villages were all semi-waterside villages. After the completion of the modern road network in 2008, high-GG villages became more numerous, and most of them overlapped with township centers.

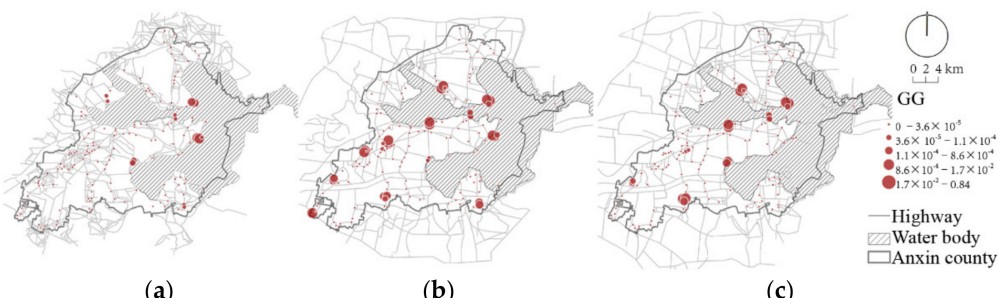

**Figure 8.** Global gravity centrality (GG) of villages in (**a**) 1964, (**b**) 1996, and (**c**) 2008.

Figure 9 shows the results for global straightness (GS), indicating the ratio of Euclidean distance to network distance between a village and all accessible villages. The smaller the value, the more physically rugged the roads. The value did not show significant changes across the studied years. However, the spatial distribution reveals that villages close to the water usually have complicated access roads. The roads of villages in the southwest area, far from the water, are nearly straight lines.

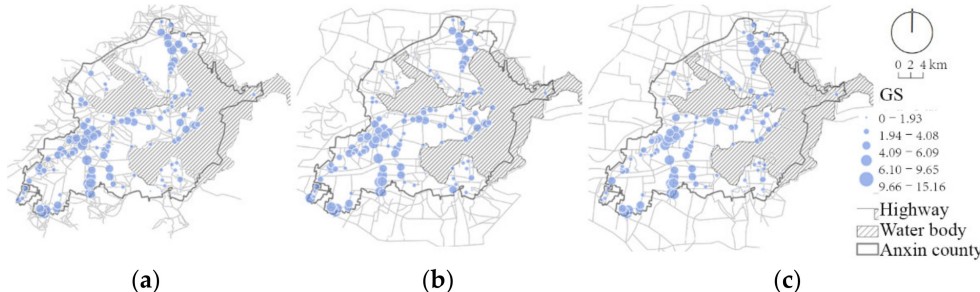

**Figure 9.** Global straightness centrality (GS) of villages in (**a**) 1964, (**b**) 1996, and (**c**) 2008.

### 3.2.2. Local Centrality Results

As shown in Figure 10, the hot spots and cold spots of local centrality were identified by Getis-Ord Gi* statistics [32]. These revealed the following:

1.  The hot spots of local reach (CR60, CR120) spread from the central plain to the north and south edge of the county border. The area around Xizhuang Village, which is along Baoding-Anxin Provincial Highway, is always a hot spot, while Liulizhuang Town is always a cold spot.
2.  The high values of local betweenness (CB60, CB120) mostly belonged to semi-waterside villages in 1964. In 1996, however, the CB values of semi-waterside villages decreased. Furthermore, in 2008, in the western region, the territory of Tongkou Town and Lao-hetou Town boasted high CB values, and they underwent a shift from a low value concentration of local intermediary centrality in 1964 to a high value concentration in 2008. This was triggered by the construction of Baoding-Anxin Provincial Highway and Gaoyang-Baoding Provincial Highway. The phenomena reflected by CB and CR have something in common, as both reflect the negative effect of water on the improvement of land transportation conditions. Villages near the water in Liulizhuang Town and Zhaili Town are always cold spots, with less accessible range and intermediacy. In contrast, major highways encourage transportation to nearby villages. The area around Xizhuang Village along the Baoding-Anxin Provincial Highway has increasingly high values for both accessibility and intermediacy.
3.  Almost all of the villages with high local closeness (CC60, CC120) are in Liulizhuang Town, where the villages form a natural group with large external connection impedance and strong internal connection due to the presence of water bodies.
4.  High values of local gravity (CG60, CG120) have a clear correlation with water, as the presence of water bodies leads to a local agglomeration of nearby villages, unlike

the dispersed distribution on the plains. These villages exert strong interaction forces on each other because of the short distances between them. This implies that semi-waterside villages that are close to a water body have the potential to affect the development of surrounding villages through their strong interaction force.

5.  Local straightness centrality (CS) is most strongly related to geographical location. The results are similar across different years.

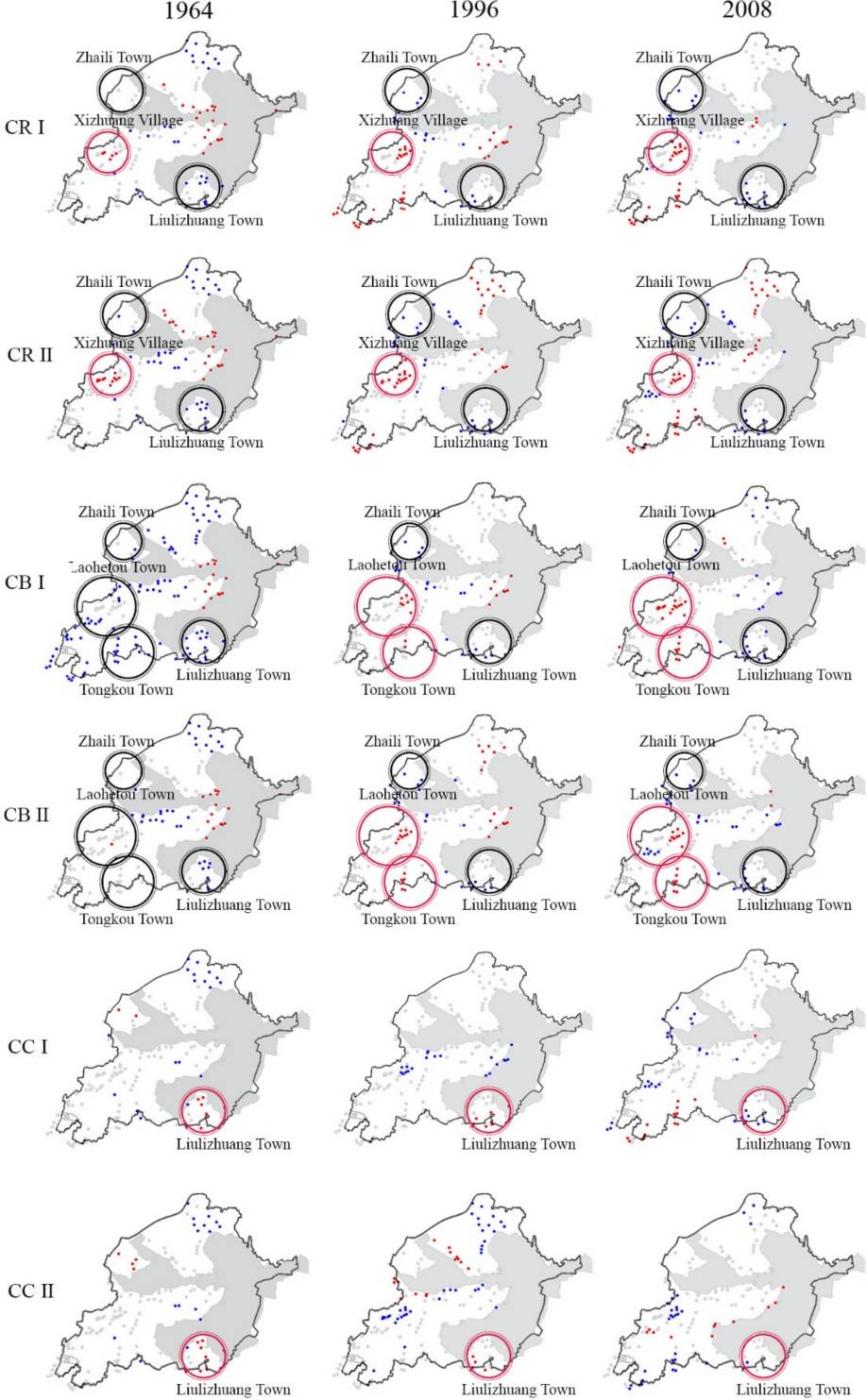

**Figure 10.** *Cont.*

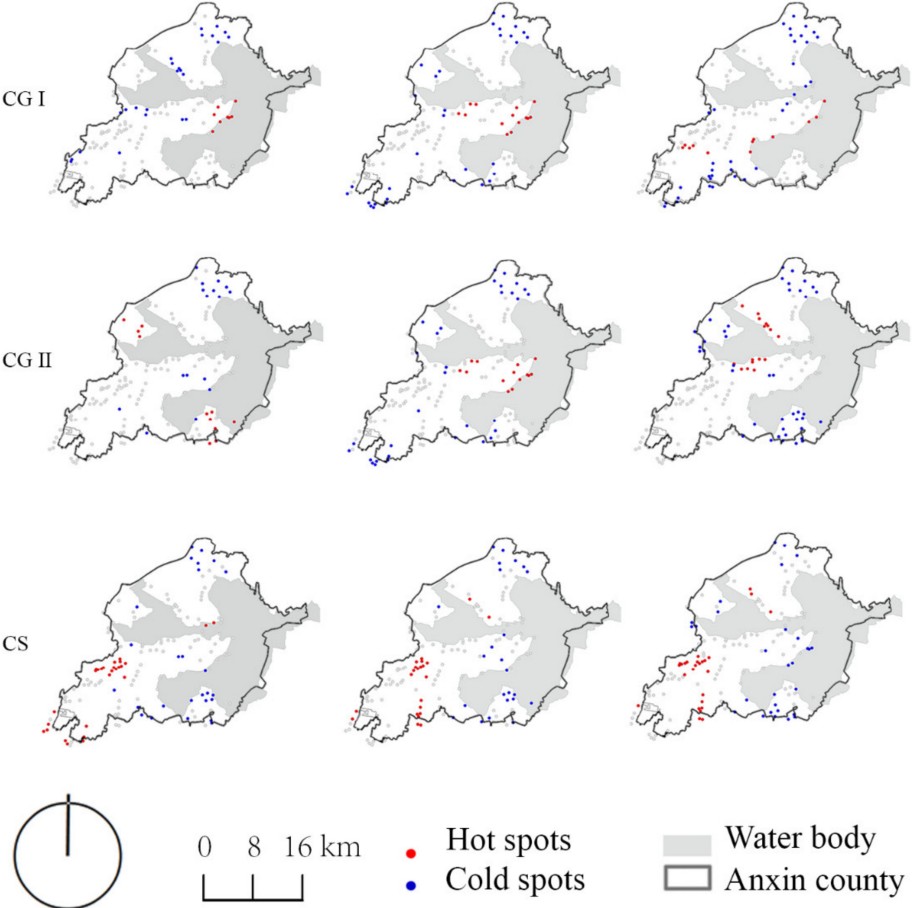

**Figure 10.** Local transport centrality indices.

### 3.2.3. Comprehensive Transport Index Results

According to the Comprehensive Transport Index (Figure 11), the *accessibility* of the villages in Anxin County changed dramatically between 1964 and 1996, but much less between 1996 and 2008. In 1996, the advantage in the Comprehensive Transport Index shifted to the north and south border of the county, such as Dawang Town in the north and Nanqing Village and Zhenzhuang Village in the south, which had an increase in transportation accessibility. Villages along the Baoding-Anxin Provincial Highway, such as the Shaopenzhuang Village–Jizhuang Village group in Dawang Town, remained at a high Comprehensive Transport Index value from 1964 to 1996. In contrast, the values for the villages along the Simen Dike and Xushui-Anxin County Highway declined from 1964 to 1996. In 2008, most high-Comprehensive Transport Index villages still maintained their advantages. In addition, the villages in Dawang Town had an even clearer advantage, while semi-waterside villages suffered a further loss. It can be concluded that high Comprehensive Transport Index levels shifted from predominantly semi-waterside villages to areas relying on the main highways.

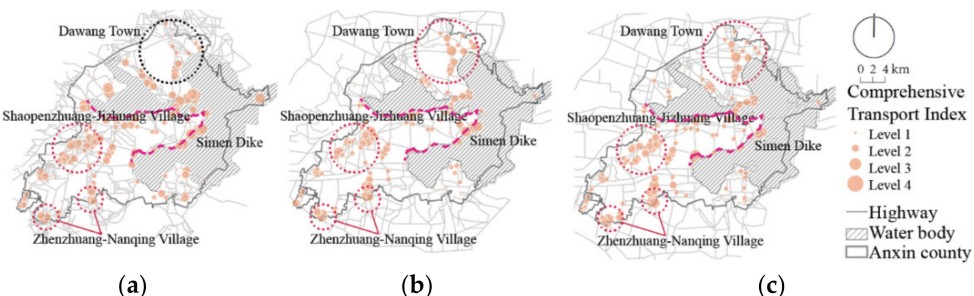

**Figure 11.** Comprehensive Transport Index of villages in (**a**) 1964, (**b**) 1996, and (**c**) 2008.

### 3.3. Village Development Index Results

The results of the Village Development Index are shown in Figure 12. The trends in village *development* (refers to socio-economic conditions) can be broadly divided into two categories. The first includes villages along the Baoding-Anxin Provincial Highway and the Xushui-Anxin County Highway, where the degree of development increased due to the road network and other township centers. The second category mainly includes villages sandwiched between the Baoding-Anxin Provincial Highway and a water body, such as the Houwumen Village–Jiuji Village area. These villages developed in the early years of the road network, but as the road system became more robust, their development slowed due to a lack of focus on connectivity with other road networks, allowing resources to be absorbed by other areas with more connectivity advantages. Interestingly, Liulizhuang Town and Zhaili Town maintain high Village Development Index values, although they have low values for most of the transport centrality indices.

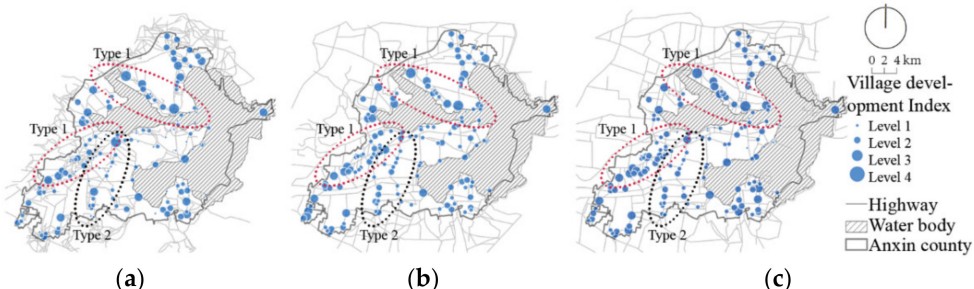

**Figure 12.** Village Development Index of villages in (**a**) 1964, (**b**) 1996, and (**c**) 2008.

### 3.4. Transport Development Index Results

We estimate the Transport Development Index by considering the Village Development Index value as node weights in a weighted centrality calculation, as shown in Figure 13. In 1964, most of the semi-waterside villages and some land villages (e.g., near Qiantingzi Village and Liuminzhuang Village along the Xushui-Anxin County Highway) had high Transport Development Index values. By 1996, the potential of many villages, as indicated by the index, had increased because of the initial construction of the modern road network. However, there are some exceptions: villages in Liulizhuang Town kept a low value. Villages along the Xushui-Anxin County Highway and Simen Dike even dropped to a lower value, indicating that semi-waterside villages lost their advantages. In 2008, only a small number of villages within Dawang Town and along the Baoding-Anxin Provincial highway, which generally had high values in the previous period, retained their high values, such as Nanliu Village and Laladi Village, while others lost their advantages. The values of the villages along the Xushui-Anxin County Highway, whose potential declined in the previous period, rebounded. Moreover, villages in Luzhuang Town, located on the southern edge, boasted high potential in all research periods.

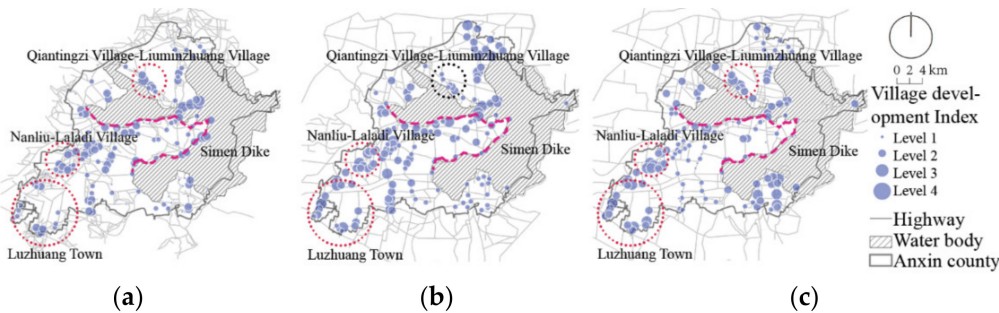

**Figure 13.** Transport Development Index of villages in (**a**) 1964, (**b**) 1996, and (**c**) 2008.

*3.5. Classification of Village Development Types*

Based on the aforementioned indices, we identified four types of villages; these are core, sub-core, connectivity villages, and sub-villages. The strategy used to classify villages into the four types showed in Table 4.

**Table 4.** Identification criteria of village types.

| Target | Type | Type Characteristics | High-Value Index | Representative Village |
|---|---|---|---|---|
| Constructing development strategy under land transportation mode | Core villages | Transport advantage, self-development, potential | Comprehensive Transport Index, Village Development Index, Transport Development Index | Laicheng |
| | Sub-core villages | Local transport advantage, self-development, potential | Local centrality dimension reduction, Development Index, Transport Development Index | Beiliucun |
| | Connectivity villages | Transport advantage, water proximity | Comprehensive Transport Index | Sanyicun |
| | Sub-villages | others | / | Zhongliucun |

The Village Development Index, Comprehensive Transport Index, and Transport Development Index are taken as the evaluation criteria for identifying the core villages. As the core villages are downstream of the town-level market and need to facilitate the surrounding villages, their land transportation accessibility and their own development condition are equally important. Similar indices, especially the transportation accessibility in a smaller range (calculated by local centralities dimensionality reduction), are also employed as criteria for identifying sub-core villages, which are complementary to core villages and require local transportation advantages and their own development. Connectivity villages are identified using the Comprehensive Transport Index, as they are a transit point in the land transportation system and require strong transportation advantages. The rest are classified as sub-villages.

All villages are categorized by the criteria above and then the results are adjusted to keep a reasonable proportion and spatial layout among four types (Figure 14). The adjustment process obeys three rules. First, for core villages and sub-core villages, their quantity should be strictly controlled, and they ought to be distributed evenly over the whole county to provide balanced market services. For instance, Laicheng Village has a high rank in all aspects, making it an appropriate core village. Sub-core villages focus on their own development and local transportation accessibility. They are chosen to complement the core villages. For example, Beiliu Village could serve as a complement to the core village catchment area on the north shore of Baiyangdian Lake. Second, connectivity villages should keep their distance from core and sub core villages, in order to maintain contact

with different village groups. For instance, Dazhaozhuang Village links the north and south lake shores; and Sanyi Village links villages in Liulizhuang Town and village groups in the west. It helps Liulizhuang Town to break the restriction of water body. Moreover, more semi-waterside villages are designated as connectivity villages in the fine-tuning process because they are intermediaries in transportation between land and water and would play an important role in modern transport systems.

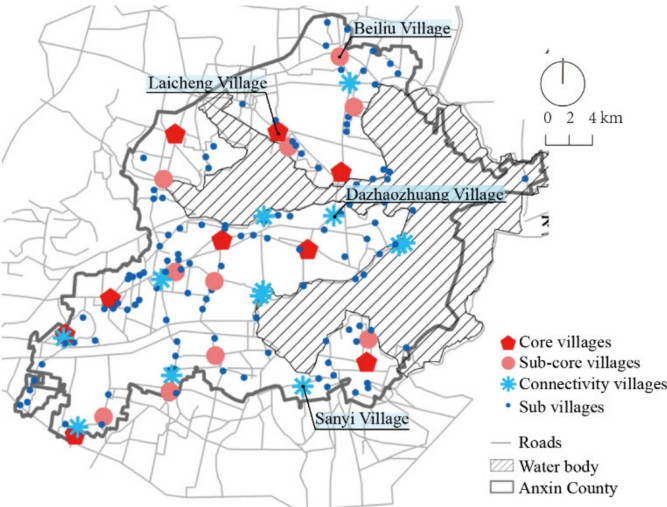

**Figure 14.** Village functionality categories in Anxin County.

## 4. Village Clusters, Dual Effects, and Transport Advantage Shifting in Anxin County

In terms of geographical proximity, nine village clusters in the county were identified, which have similar transportation accessibility and potential (Figure 15).

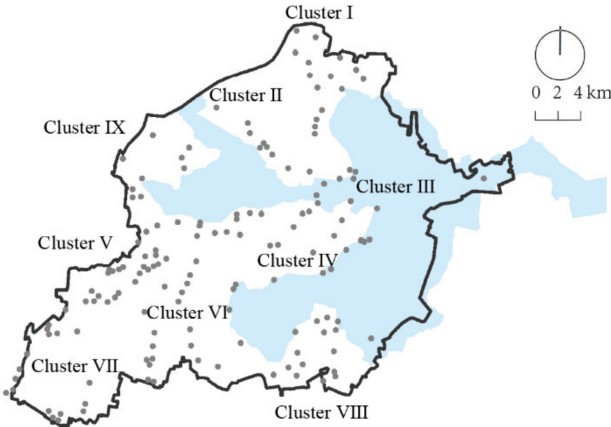

**Figure 15.** Village clusters in Anxin County.

Cluster I: This cluster pioneered in developing when the modern road network was first built and consequently gained a high transportation accessibility advantage. It benefited from the proximity of Rongcheng County and the Rongcheng-Lixian Provincial Highway. Its potential rose at first but then fell again, due to the construction of the northern section of the Rongcheng-Lixian Provincial Highway, which added a route to Rongcheng County and dispersed the flow.

Cluster II: This cluster consists of semi-waterside villages, whose transportation accessibility advantage declined when the modern road network was first built but rose again after the re-routing of Xushui-Anxin County Highway, which increased connectivity with other road networks. The transportation accessibility advantage of this cluster is more reflected in their attraction of surrounding villages. This cluster is driven by both

water transportation and the Xushui-Anxin County Highway, and thus most villages in the cluster enjoy both high transportation *accessibility* and *potential*.

Cluster III: This cluster is near the periphery of Anxin County and also consists of semi-waterside villages, which have a strong transportation accessibility advantage but was weakened slightly after the construction of the modern road network. However, because the cluster is close to the county administrative center, these villages enjoy high potential and retain a certain transportation accessibility advantage.

Cluster IV: This cluster includes several semi-waterside villages along the Simen Dike and some inland villages. The former lost their advantage in transportation accessibility due to the construction of the modern road network. The latter kept a low level of transportation accessibility because of their sparse roads. The cluster should be prioritized for optimization because of the villages' low *potential* if the current situation remains unchanged.

Cluster V: The villages of this cluster are located along Baoding-Anxin Provincial Highway. Thus, they have both high transportation accessibility and *potential*.

Cluster VI: This cluster is along the southern section of Rongcheng-Lixian Provincial Highway and thus has a certain transportation accessibility advantage. However, the linear arrangement of villages leads to some isolation from other villages and low potential. It should be prioritized for optimization, like Cluster IV.

Cluster VII: This cluster is close to the Gaoyang-Baoding Provincial Highway. Villages in this cluster have shown a rapid rise in transportation accessibility since the completion of the modern road network. This cluster also has considerable *potential*.

Cluster VIII and Cluster IX are Liulizhuang Town and Zhaili Town, respectively, which are in a similar situation. Surrounded by water on two sides, their transportation is characterized by strong internal cohesion but weak external links. However, benefiting from the convenience brought by the earlier water bodies, they have higher degrees of development and a higher potential for development. This description above of the nine clusters reveals the spatial imbalance in development opportunities among waterside villages in Anxin County. Some clusters should seek support from more powerful clusters.

Imbalance among clusters shows the dual influence exerted by water bodies on waterside villages. The water limits the external connection of the villages via land transportation, such as Zhaili Town and Liulizhuang Town. Their centrality indices show a strong cohesion but weak external connectivity. In addition, the positive effect of the water body shows a path-dependent effect. Semi-waterside villages displayed both a high development index and transportation accessibility advantage in 1964. Until it was restored by the completion of the modern road network in 2008, these villages lost their accessibility advantage because the connection between the road network and body of water was weakened. However, they still kept a high-level self-development, which shows an inertia derived from water. Zhaili Town and Liulizhuang Town are typical cases. There is further evidence which proved that such dual effect is disappearing, while building a land transportation-oriented developing mode is becoming necessary. In previous stage of the study, we analyzed the relationship between spatial accessibility and the development degree of villages by correlation and regression analysis. As mentioned in Section 1 of this paper, the result showed that the correlation was not significant in 1964 and 1996 but weakly significant in 2008. It may because that in the water transportation era, some villages along the water usually developed into market towns and, consequently, had higher populations or higher per capita incomes than other villages, although they had weak road accessibility. The correlation result indicates that this mismatching continued until 1996, then weakened in 2008, and has gradually faded since. It is a reasonable explanation that if more roads were be built around the semi-waterside villages, they would easily gain high a *potential* through the land transport system over the water-driven inertia. Moreover, because of their high gravity centrality index, the path-dependent positive effect exerted by water can be diffused to other villages according to a build-up of roads, which would further improve the regional development level.

Aligned with previous studies, the effect was exerted not only by the body of water but even more strongly by the main road network. Clusters close to the main road showed rapid development, except Cluster VI, which proved that the southern section of Rongcheng-Lixian Provincial Highway had a weaker driving effect. Therefore, by promoting road construction around semi-waterside villages and strengthening the connectivity of the southern section of the Rongcheng-Lixian Provincial Highway, the impacts of water and main roads can be better utilized for village development.

## 5. Conclusions

This paper measures the evolution of village land transportation accessibility and *potential* since the beginning of the shift in transport mode in Anxin County by analyzing the spatial network and its centrality weighted by the Village Development Index. Based on this analysis, a development strategy adapted to the modern transport system is established for waterside villages in Anxin County.

The evolution of the road network system indicates that the network changed from disorder to regularity then to abundance in morphology from 1964 to 2008, when a modern highway system was established to connect each village. In this process, road networks' connection with water was weakened.

To answer the first question, temporal changes in an *accessibility* advantage reflect the redistribution of transport resources during the construction of the land transportation system in the study area. The dominant process switches from cohesiveness with water bodies to an even distribution then to an aggregated distribution dominated by the road network. In terms of village potential, in 1964, when the modern road network system was not yet built, villages either close to water bodies or to the road network had high potential. In contrast, in 2008, when the modern road network was completed, villages that benefitted from proximity to both the road network and the surrounding villages gained high potential. Thus, it can be inferred that to meet the two requirements above, a hierarchical clustered structure of waterside villages should be built, relying on the main road network to meet the new need for a modern transport system.

Next, to answer the second question, we first proposed a land transportation-oriented hierarchical spatial structure to adapt to the modern transport system, including core villages, sub-core villages, connectivity villages, and sub-villages. In order to save resources, governments could provide different development strategies for them. Core or sub core village could be given priority to more construction facilities, so that surrounding lower-grade villages can also have access to services and facilities provided by them via connectivity villages. Moreover, the hierarchy can support the upcoming "Village Relocation and Combination Planning" for Baiyangdian Lake Area (proposed in the *Baiyangdian Ecological Environment Management and Protection Planning*). Population, size, and transport condition are key factors to identify villages with a high residential field and power, who hold more attraction for households to move to. In the hierarchy village structure proposed in Section 3.5, core villages and sub-core villages have advantages in these three factors [33]. Thus, they should be the primary choice as destinations for population relocation while having the top priority for investment. The strategies of developing connectivity villages need to focus on strengthening their links to the core villages. Employing semi-waterside villages as connectivity villages facilitates the integration of water and land transportation modes to produce a greater positive effect. In addition, because sub-villages are not outstanding in all parameters, in decision-making they can be merged with core or sub-core villages into batches in order to increase the efficiency of land use. Secondly, by discovering the temporal law of *accessibility* and *potential*, we find that promoting road construction around semi-waterside villages and strengthening the connectivity of the southern section of the Rongcheng-Lixian Provincial Highway will effectively enhance the efficiency of the modern transport system.

This study has some limitations because of the lack of data from the past decade to support an even finer-grained study. From a methodological perspective, villages that are

located entirely in water bodies were not investigated or considered, which may slightly affect the accuracy of the *accessibility* if integrating both water and land transportation modes. Despite this deficiency, the ability of the centrality indices to guide the transition of waterside villages to land transportation modes was demonstrated. By taking transport accessibility and socioeconomic development into consideration, the centrality index can not only interpret the evolution of transportation but also measure the potential of villages under the land transportation mode. Consequently, it contributes to building a new structural system of waterside villages to better develop modern transport system. Furthermore, it can also guide decision-making in issues of local government concern.

**Author Contributions:** C.W. Conceptualization; methodology; software; formal analysis; writing–original draft preparation; visualization. J.H. writing–review and editing; supervision; project administration; funding acquisition. All authors have read and agreed to the published version of the manuscript.

**Funding:** This research was Supported by the Stable Support Programme for Higher Education by Shenzhen R&D Funds, Grant No. GXWD20201230155427003-20200803172955008.

**Institutional Review Board Statement:** Not applicable.

**Informed Consent Statement:** Not applicable.

**Data Availability Statement:** Data sharing not applicable.

**Conflicts of Interest:** The authors declare no conflict of interest.

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
