# Peer review of "The Transformation and Development Strategy of Waterside Villages through Transport System Reconstruction: A Case Study of Anxin County, Hebei Province, China"

_applsci, doi:10.3390/app12126142_

Round 1

Reviewer 1 Report

As a review of the development of the transport system in the study area and the impacts of the developed land roads/highway on the semi-water villages and land villages, this paper has some value, and its ambitions are good. The paper presented some valuable data that were collected and calculated for the accessibility of the road system in the area.

However, there are two main problems in its research methods. First is the confusion of the scales of the study cases.  About 10 diverse townships in the area were taken as case studies and were classified into two types: those close to the water or not close to the water.  Within each township, however, there are also numerous land villages and semi-water villages. The calculations of spatial accessibility and socio-economic condition of those areas are based on the data for villages rather than those for townships. However, the discussions of the results of calculations were keep changing between villages and townships but those are not interchangeable concepts.

It is not very clear about the relationships between the calculated spatial accessibility of villages, based on road conditions, and the calculations of villages’ socio-economic conditions, based on numbers of population, per capita income, area, and area growth of the place. Did the villages develop more than other villages because they were closer to the transport nodes or the situations were much more complicated than this simple reason. The categories of four village types proposed by authors in the study seem to indicate that was the case. However, if a village has a lower passing speed (in its accessibility calculation), does that mean the roads in the village are less developed to drive through, or would the reason being that there were attractions, such as developed markets, in the village that keep people staying there longer than in other villages.

This is an ambitious project, but research methods are over-simplified to analyse the development of traditional waterside villages and modern transport system in such a large and diverse area, therefore the conclusion is less convincing.

Reviewer 2 Report

With the present manuscript on the impact of water conservation policies on the transportation ways in waterside villages, the authors get to an interesting conclusion in the form of a potential proposal for a reference transportation model.

Of course, the critical limit of such an approach is the difficulty of getting reliable data from the past decade to support an even fine-grained study. As also mentioned by the authors in line 516.

Although the manuscript's limits are clearly introduced with a lucid background and research questions, a better understanding of the research design requires elucidating a couple of issues.

One is about the resolution and parameters of data selected for examining the changes in land transportation accessibility and potential in the three chosen periods, used to map the road infrastructure, calibrate the images and scan maps (declassified CORONA, Landsat and Landsat OLI). Reading from section 2.2, it is unclear what the actual parameters are. This is a critical aspect on which the overall quality of the whole method and final results depend. My suggestion is to add a descriptive table with data parameters.

The second one is about the calculation of multiple centrality indices. In particular, on the chosen search radius as travel times to study the road centrality at different scales. How the time thresholds of 60, 120, and 1000 minutes have been selected?

Other minor issues are as follow:

  • English grammar and style should be checked. As an example Line 22 - 23 looks incomplete "These four types form a spatial structure THAT can not only BE adapted to the modern transport system..." Finally, I might be mistaken or biased by my western-eye, but I do not understand the study area in Figure 1. The wider area containing the studied County is a city? I mean, is ok the City of Baoding is made of towns and administrative villages?

Reviewer 3 Report

37-38 incorrect sentence: A modern transport system consists of strong land transportation and weak water transportation had formed.

47-48 For example, the waterborne-transportation-dominated structure and corresponding spatial layout of the villages in the Lixia River region of  northern Jiangsu Province (in east China) were rapidly reconfigured by the strong intervention of the land transport system [7]. Explain how were they reconfigured.

89 - 92 incorrect sentence: The first is accessibility, refers to the ability of a village to be reached in road network; the second is potential energy, simply called potential, is defined as the capability of a village develop  into an important node in the road network, is a comprehensive evaluation of accessibility and socio-economic condition.

502 -  506, Please explain more this: Besides, the hierarchy can support the upcoming “Village Relocation and Combination Planning” in Baiyangdian Lake Area (proposed in the Baiyangdian Ecological Environment Management and Protection Planning). Based on the revealed development mechanism, core villages and sub-core villages should be the primary choice as destinations for population relocation and have top priority for investment.

The text required minor spell check.

The paper focuses mostly on road transport infrastructure and further development of villages according to the types of villages derived during research. However other development aspects such as land use, or the presence of industry are missing. The article is fine and the research uses the usual methodology, but there is no assessment of the potential of waterborne transport. Is it completely extinct in this area? Does it have any potential?

Round 2

Reviewer 1 Report

Authors provided clear clarification and explanations to two questions raised in the review in their Response to Reviewer 1. As a result, two changes were made in the paper. However, the explanation to the second question should have been included in the paper to define the scope of this particular study clearly, and point out what outcomes are referred from previous study and what are not included in the discussions in this particular paper. 

Reviewer 3 Report

No further comments.

Author Response

The co-anthors and I would like to thank you for the time and effort spent in reviewing the manuscript.